# Molecular characterization of the intact mouse muscle spindle using a multi-omics approach

**Bavat Bornstein[1]*[†], Lia Heinemann-Yerushalmi[1][†], Sharon Krief[1], Ruth Adler[1], Bareket Dassa[2], Dena Leshkowitz[2], Minchul Kim[3,4], Guy Bewick[5], Robert W Banks[6], Elazar Zelzer[1]***

[1]Department of Molecular Genetics, Weizmann Institute of Science, Rehovot, Israel; [2]Bioinformatics Unit, Department of Life Sciences Core Facilities, Weizmann Institute of Science, Rehovot, Israel; [3]Developmental Biology/Signal Transduction, Max Delbrueck Center for Molecular Medicine, Berlin, Germany; [4]Team of syncytial cell biology, Institut de Génétique et de Biologie Moléculaire et Cellulaire (IGBMC), Illkirch, France; [5]Institute of Medical Sciences, University of Aberdeen, Aberdeen, United Kingdom; [6]Department of Biosciences, Durham University, Durham, United Kingdom

**\*For correspondence:**
bavat.bornstein@weizmann.ac.il (BB);
eli.zelzer@weizmann.ac.il (EZ)

[†]These authors contributed equally to this work

**Competing interest:** The authors declare that no competing interests exist.

**Abstract** The proprioceptive system is essential for the control of coordinated movement, posture, and skeletal integrity. The sense of proprioception is produced in the brain using peripheral sensory input from receptors such as the muscle spindle, which detects changes in the length of skeletal muscles. Despite its importance, the molecular composition of the muscle spindle is largely unknown. In this study, we generated comprehensive transcriptomic and proteomic datasets of the entire muscle spindle isolated from the murine deep masseter muscle. We then associated differentially expressed genes with the various tissues composing the spindle using bioinformatic analysis. Immunostaining verified these predictions, thus establishing new markers for the different spindle tissues. Utilizing these markers, we identified the differentiation stages the spindle capsule cells undergo during development. Together, these findings provide comprehensive molecular characterization of the intact spindle as well as new tools to study its development and function in health and disease.

## Editor's evaluation

This works provides a valuable and comprehensive description of the molecular composition of the different compartments of the muscle spindle. The authors combine convincing transcriptomic, proteomic and imaging approaches to provide the field with new tools for dissecting the development and function of the muscle spindle. This manuscript is of interest for a broad spectrum of researchers working on the nervous and muscular systems.

## Introduction

The proprioceptive system is essential for controlling coordinated movement and posture. Proprioceptive information is produced by specialized mechanosensory organs located in muscles, tendons, and joints, which detect the stretch, tension, and force experienced by the muscles. This information is then transferred to the central nervous system, where input from populations of proprioceptive

neurons is integrated to generate the sense of position and movement of limb and trunk (*Kiehn, 2016*; *Proske and Gandevia, 2012*; *Sherrington, 1907*).

In mammalians, the muscle spindle is one of the main proprioceptive mechanosensory organs (*Proske and Gandevia, 2012*). The spindle is composed of specialized muscle fibers, termed intrafusal fibers, which are innervated by proprioceptive sensory neurons at their central region and by γ-moto-neurons at their polar ends. This structure is partially isolated from its surroundings by a capsule rich in extracellular matrix (ECM) that is secreted by capsule cells (*Bewick and Banks, 2014*; *Kröger and Watkins, 2021*). The development of the muscle spindle starts in utero and continues postnatally. This process is initiated when sensory neuron afferents contact immature myofibers and induce their differentiation (*Hippenmeyer et al., 2002*). Nonetheless, the molecular events that regulate spindle development are largely unknown.

Previous work from our lab demonstrated that the proprioceptive system regulates several aspects of musculoskeletal development and function and that impaired proprioceptive signaling causes musculoskeletal pathology (*Assaraf et al., 2020*; *Blecher et al., 2017a*; *Blecher et al., 2017b*; *Born-stein et al., 2021*). These findings significantly increase the importance of the proprioceptive system and emphasize the need to understand the molecular mechanisms underlying its development and function.

In recent years, attempts have been made to uncover the molecular composition of the spindle. Single-cell RNA analysis of dorsal root ganglion (DRG) proprioceptive neurons revealed their molecular diversity, subdividing these neurons to five to eight subgroups and identifying specific markers for the proprioceptive neurons composing muscle spindle (*Oliver et al., 2021*; *Wu et al., 2021*). Single-nucleus transcriptomic analysis of spinal cord motoneurons identified the transcriptional profile of γ-motoneurons (*Blum et al., 2021*). These studies greatly advance our molecular understanding of proprioceptive and γ-motoneurons. However, these datasets contain RNA transcripts expressed at the neuron cell body and nucleus, missing information on transcript expression at the terminal ends. Because gene expression in neurons is regulated locally (*Holt and Schuman, 2013*), information on localized expression in specialized sensory regions is central to our understanding of spindle biology. Another recent advance in spindle research was made by single-nucleus sequencing of intra-fusal fibers, which identified six nuclear compartments within these fibers (*Kim et al., 2020*). However, the molecular compositions of intrafusal versus extrafusal fibers were not compared. Another missing piece of information is the molecular composition of the spindle capsule.

To address these knowledge gaps and to provide comprehensive gene expression profiles of the muscle spindle, we took a holistic approach and performed transcriptomic and proteomic analyses on intact spindles. We then analyzed the obtained datasets to identify novel markers for the different spindle tissues. Finally, we used these markers to study postnatal development and maturation of capsule cells.

## Results

### Transcriptomic analysis provides a molecular characterization of the intact muscle spindle

The molecular details of the muscle spindle and the various tissues that compose it are largely missing. To provide a comprehensive molecular characterization of this complex organ, we isolated intact muscle spindles and subjected them to transcriptomic and proteomic analyses. Two major hurdles in studying the muscle spindle are its small size and infrequent occurrence within different muscles. To overcome these hurdles, we isolated intact spindles from the deep masseter muscle, which is known to be rich in these receptors (*Lennartsson, 1980*). To recognize the spindles inside the masseter belly, we marked them genetically by crossing *Piezo2*$^{EGFP-IRES-Cre}$ deleter mice with a *Rosa26*$^{tdTomato}$ reporter line (*Madisen et al., 2010*; *Woo et al., 2014*). *Piezo2* was previously shown to be expressed by proprioceptive sensory neurons in the spindle (*Woo et al., 2015*). To verify the specific expression of tdTomato in spindles of *Piezo2*$^{EGFP-IRES-Cre}$;*Rosa26*$^{tdTomato}$ mice, we first examined sections through the masseter and found a strong signal in the spindles (*Figure 1A*). We then compared *Piezo2* expression, as indicated by GFP signal, to the expression of tdTomato (*Figure 1—figure supplement 1*). As reported previously (*Woo et al., 2015*), GFP was detected in sensory neuron soma in the DRG and at the proprioceptive neuron endings (*Figure 1—figure supplement 1*). Interestingly, while tdTomato

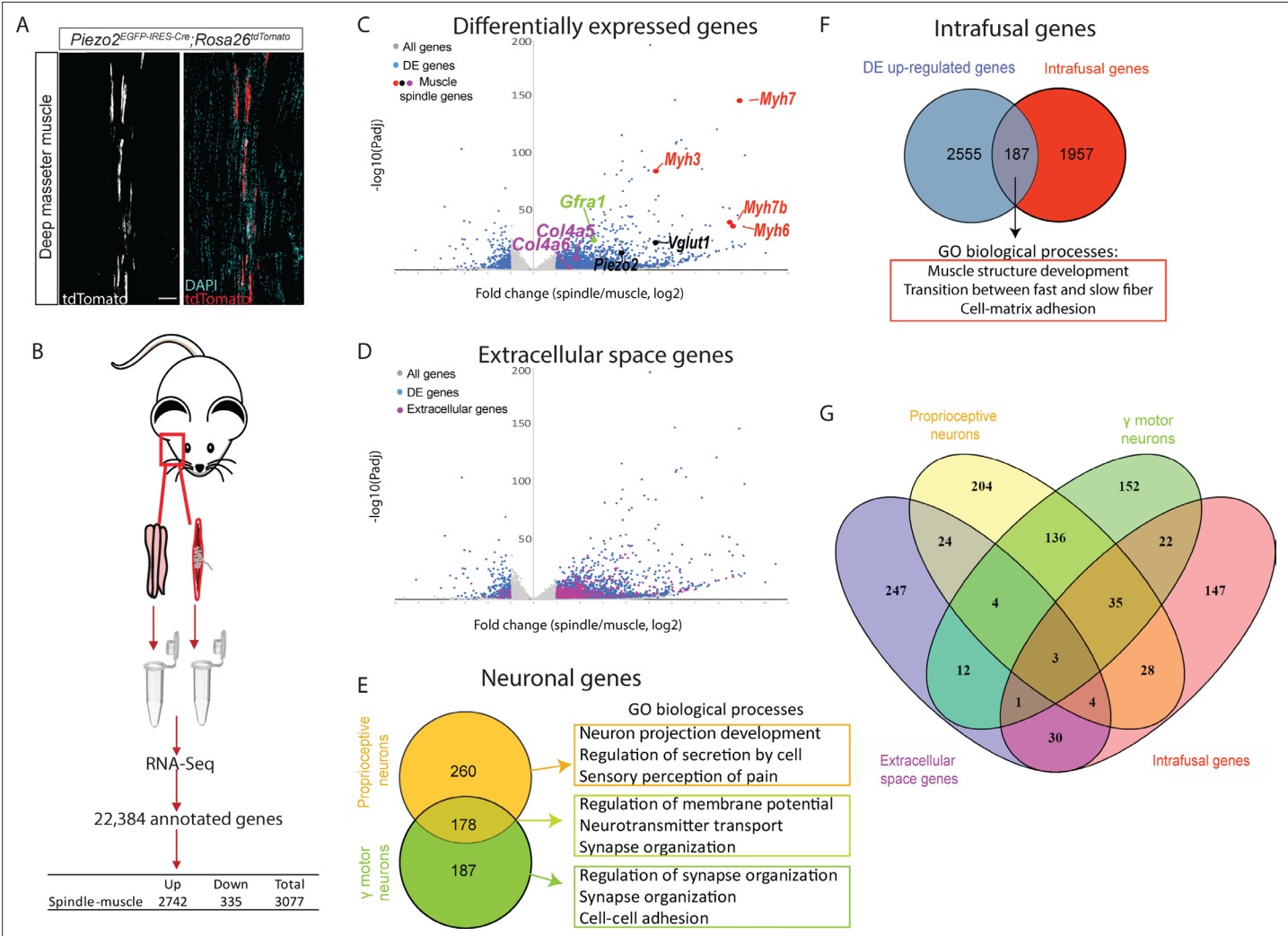

**Figure 1.** Transcriptomic analysis of intact muscle spindles identified genes expressed in the different spindle tissues. (**A**) Confocal images of longitudinal sections of the deep masseter muscle of adult (>P90) *Piezo2*$^{EGFP-IRES-Cre}$;*Rosa26*$^{tdTomato}$ mice. The expression of TdTomato shows the abundance of muscle spindles in this muscle. White (left) and red, tdTomato; cyan, DAPI; scale bar represents 50 μm. (**B**) Schematic representation of sample isolation and sequencing. Bulk transcriptomic analysis was performed on intact muscle spindles and adjacent extrafusal muscle fibers (muscle). The table contains the number of genes that were differentially expressed between spindle and muscle samples. (**C**) Volcano plot depicting differentially expressed (DE) genes between spindle and muscle samples. Gray dots represent all detected genes; blue dots represent DE genes. Other colored dots indicate genes known to be expressed in intrafusal fibers (red), proprioceptive neurons (black), γ-motoneurons (green), and muscle spindle capsule (magenta). Y-axis denotes −log10 (p-values), whereas X-axis shows log2 fold change values. (**D**) Volcano plot depicting DE genes between spindle and muscle samples. Gray dots represent all detected genes; blue dots represent DE genes; magenta represent DE genes that are located at the extracellular. Y-axis denotes −log10 (p-values), whereas X-axis shows log2 fold change values. (**E**) Left: A Venn diagram showing DE genes potentially expressed by proprioceptive neurons (orange) and γ-motoneurons (green). The overlap between the two datasets is marked by light green. Right: Gene ontology (GO) analysis for enriched biological processes in each dataset using Metascape (see also **Supplementary file 4**, **Supplementary file 5**). (**F**) A Venn diagram showing the overlap between upregulated genes in our analysis (blue) and intrafusal genes previously reported by **Kim et al., 2020** (red). Below are the most enriched biological processes in the shared genes, as indicated by GO analysis using Metascape (see also **Supplementary file 6**; **Supplementary file 7**). (**G**) A Venn diagram of the four groups of DE genes displayed in D–F, namely genes associated with the extracellular space (magenta), proprioceptive neurons (yellow), γ-motoneurons (green), and intrafusal fibers (red).

The online version of this article includes the following figure supplement(s) for figure 1:

**Figure supplement 1.** Expression of tdTomato in *Piezo2*$^{EGFP-IRES-Cre}$;*Rosa26*$^{tdTomato}$ mouse.

**Figure supplement 2.** The RNA sequencing (RNA-seq) data contain neuronal transcripts.

expression was not detected in the sensory endings, it was expressed in DRG sensory neurons, intra-fusal muscle fibers, and capsule cells (*Figure 1—figure supplement 1*). The expression of tdTomato in the DRG but not in the neuron endings was probably due to lack of transport of the reporter to the terminal ends. Moreover, the expression of tdTomato, but not GFP, in intrafusal fibers and capsule cells suggests that *Piezo2* was expressed in the progenitors of these lineages. Nonetheless, this broad tdTomato expression enabled us to manually dissect out the intact spindle from the masseter belly. From each mouse, we obtained intact muscle spindles as well as extrafusal muscles fibers adjacent to the spindle, referred to as muscle. The muscle samples were used as a reference tissue by which to identify spindle-specific genes and to eliminate possible contamination with extrafusal fibers.

Next, we performed bulk RNA-sequencing (RNA-seq) of spindle and muscle samples (*Figure 1B*). Principal components analysis (PCA) indicated different transcriptional states for the two tissue types (*Figure 1—figure supplement 2A*). Differential expression analysis identified over 3000 genes that were differentially expressed between spindle and muscle samples (*Figure 1C*; *Supplementary file 1*; see Materials and methods for details). To verify these results, we searched our dataset for the expression of known muscle spindle markers. We found that markers for intrafusal fibers (*Myh3*, *Myh6*, *Myh7*, and *Myh7b*) (*Lee et al., 2019*; *Schiaffino et al., 2015*; *Soukup et al., 1995*; *Walro and Kucera, 1999*), for proprioceptive sensory neurons (*Piezo2* and *Slc17a7*, also known as *Vglut1*) (*Bewick et al., 2005*; *Woo et al., 2015*), as well as for ECM (collagen type IV; *Sanes, 1982*) and γ-motoneurons (*Gfrα1*; *Shneider et al., 2009*), were upregulated in spindles relative to muscle (*Figure 1C*). To gain more information about upregulated genes, we performed gene ontology (GO) enrichment analysis using the Metascape web tool (https://metascape.org/). We identified enrichment for cellular compartments such as 'external encapsulating structure', 'ECM', 'synaptic membrane', 'axon', and 'axon terminus' (*Figure 1—figure supplement 2B*; *Supplementary file 2*), suggesting that our data include RNA transcripts that originate from capsule cells and neurons. Taken together, the finding of known markers for muscle spindle tissues and the GO analysis results suggest that we successfully obtained comprehensive transcriptomic data from the entire spindle.

To better understand the muscle spindle transcriptome, we analyzed our data for genes associated with the various spindle tissues, starting with extracellular genes. Ingenuity pathway analysis showed that 325 differentially upregulated genes were located to the extracellular space (*Figure 1D*, *Supplementary file 3*). This list contained many matrix proteins, such as collagens and matrix metalloproteinases, as well as signaling molecules, such as of the Wnt and BMP pathways. Because the spindle capsule contains ECM components, the genes in this list are likely associated with the capsule.

Muscle spindles are innervated by proprioceptive sensory neurons and γ-motoneurons (*Proske and Gandevia, 2012*). To identify RNA transcripts that could derive from these two neuron types, we performed ranked gene set enrichment analysis (GSEA) to compare between differentially expressed genes that were upregulated in spindle samples and the expression profiles of isolated DRG proprioceptive neurons (*Zheng et al., 2019*) and γ-motoneurons (*Blum et al., 2021*). This analysis yielded two lists of 438 genes that are also highly expressed by proprioceptive neurons and 365 genes highly expressed in γ-motoneurons (*Figure 1E*, *Figure 1—figure supplement 2C and D*, *Supplementary file 4*). Comparison between the two lists revealed 178 shared genes (*Figure 1E*, *Supplementary file 4*). The remaining 260 proprioceptive neuron genes and 187 γ-motoneuron genes represent RNA transcripts that might be expressed locally at the proprioceptive and γ-motoneuron endings, respectively.

To gain more information about these sets of neuronal genes, we performed GO enrichment analysis for biological processes using Metascape (*Figure 1—figure supplement 2E*, *Supplementary file 5*). We found that while γ-motoneuron genes were enriched for processes related to synapse organization, proprioceptive neurons genes were related to neuron differentiation and development (*Figure 1E*). Furthermore, we found strong enrichment for processes such as synapse organization and trans-synaptic signaling in the 178 genes that were shared between the two neuron types. This result is consistent with previous findings that the peripheral endings of proprioceptive neurons have a number of synapse-like structural components, including synaptic-like vesicles, synapsins, and synaptobrevin/VAMP (*Bewick et al., 2005*; *Bewick and Banks, 2014*; *Than et al., 2021*; *Zhang et al., 2014*).

Proprioceptive sensory neurons can be subdivided into three types of afferent fibers, namely Ia and II, which innervate muscle spindles, and Ib, which innervates the Golgi tendon organ (GTO) (*Proske and Gandevia, 2012*). To study the specificity of the spindle afferent genes we identified,

we examined the overlap between our list of 260 potential proprioceptive neuron genes and markers for the three proprioceptive neurons subtypes identified by *Wu et al., 2021*. While we found many genes that are common to all subtypes, 22 genes exclusively overlapped with type Ia neurons, 45 genes with type II neurons, and 2 genes with both (*Figure 1—figure supplement 2F*; *Supplementary file 4*). These results suggest that these 69 genes are expressed by muscle spindle afferents and not by GTO afferents. To explore the specificity of our potential γ-motoneuron genes, we sought to filter out genes that are expressed in α-motor and proprioceptive neurons. For that, we generated a list of α-motoneurons genes by performing ranked GSEA on our data using published expression profiles of these neurons (*Blum et al., 2021*). Then, we compared between the three lists of neuronal genes, that is, those associated with γ-motoneuron, α-motoneurons, and proprioceptive neurons, and found a large overlap between the three lists (*Figure 1—figure supplement 2G*). Nonetheless, we identified 40 spindle genes that are specific to γ-motoneuron (*Figure 1—figure supplement 2G*; *Supplementary file 4*) and are, therefore, potential markers for these neurons.

Finally, to identify RNA transcripts unique to intrafusal fibers, we compared the differentially expressed upregulated genes to a previously reported expression profile of intrafusal fibers (*Kim et al., 2020*). Results showed an overlap of 187 genes between the two dataset (*Figure 1F*; *Supplementary file 6*). GO analysis on these intrafusal genes identified biological processes such as 'muscle structure development', 'transition between fast and slow fiber', and 'cell-matrix adhesion' (*Figure 1F*; *Figure 1—figure supplement 2H*, *Supplementary file 7*).

Overall, we have established a database of 2742 genes that are upregulated in muscle spindle relative to adjacent extrafusal fibers. We have demonstrated that this database includes RNA transcripts

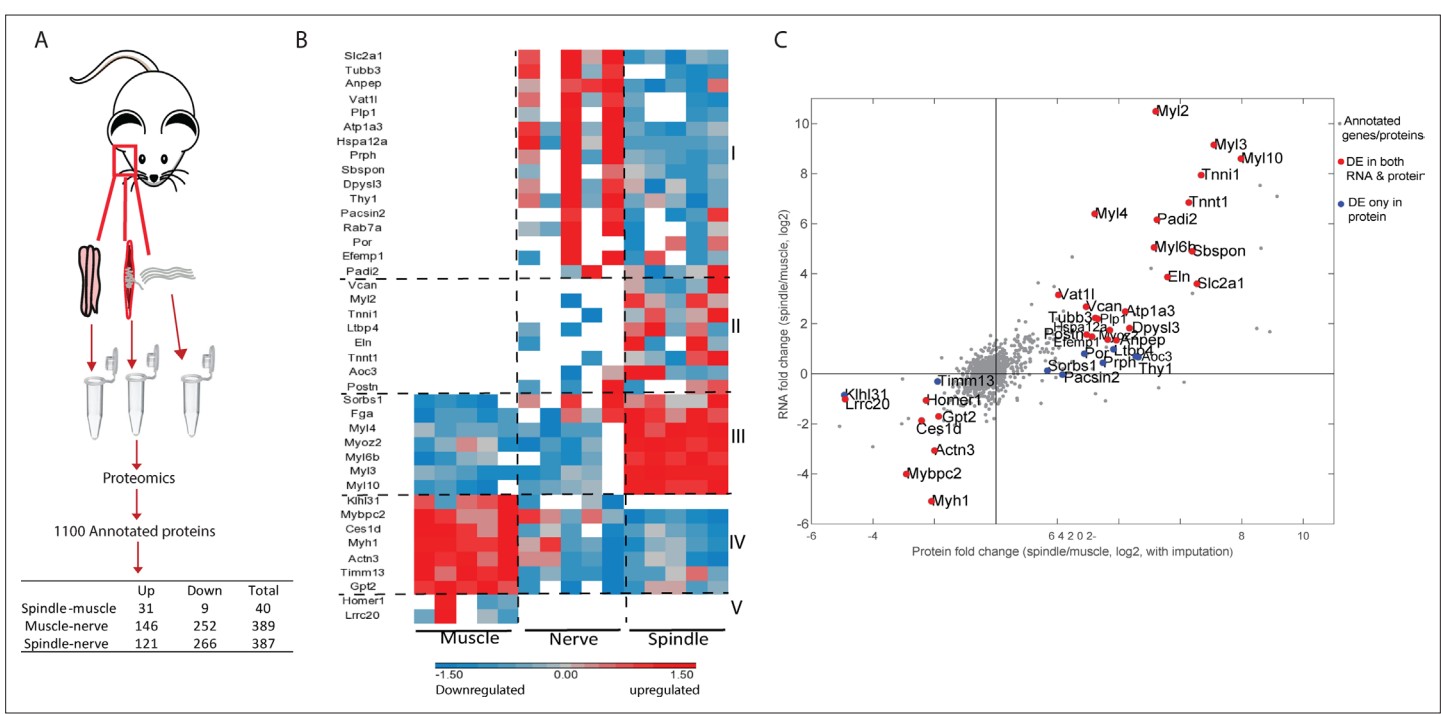

**Figure 2.** Proteomic analysis of intact muscle spindle identified proteins expressed in its constituent tissues. (**A**) Schematic representation of the analyzed samples. Proteomic analysis was performed on intact muscle spindle, extrafusal muscle fibers (muscle), and nerve fibers deprived of their nerve termini and cell bodies (nerve). (**B**) Heatmap showing clustering of the differentially expressed proteins between muscle spindle and extrafusal fibers. Each horizontal line denotes the relative expression of a single protein (log2-transformed LFQ intensities with row standardization; proteins not detected are in white). Cluster numbers are indicated by Roman letters on the right. (**C**) Scatter plot showing the correlation between the fold changes of spindle-muscle differentially expressed (DE) proteins and RNA. The X-axis indicates log2 fold change values for proteins, whereas the Y-axis shows the log2 fold change values for transcripts (shown in *Figure 1*). Gray dots represent all genes and proteins detected, red dots represent DE molecules at both RNA and protein levels, blue dots represent DE proteins only. DE protein symbols are shown on the plot.

The online version of this article includes the following figure supplement(s) for figure 2:

**Figure supplement 1.** Proteomic analysis of muscle spindles and adjacent muscle and nerve.

from all tissues composing the spindle, including intrafusal fibers, neuronal tissues, and capsule cells (*Figure 1G*), thereby providing molecular characterization of the entire organ.

## Proteomic analysis identified potential markers for the different tissues that compose the muscle spindle

Next, we performed proteomic analysis on intact muscle spindles and adjacent extrafusal fibers isolated from the deep masseter muscle. Because we expected some of the differentially expressed proteins to originate in the neuronal component of the spindle, we also analyzed nerve fibers from the masseter without their nerve termini and cell bodies (nerve; *Figure 2A*). PCA of the obtained proteomic data revealed three separate populations (*Figure 2—figure supplement 1A*), indicating that muscle spindles, extrafusal fibers, and nerve fibers display different protein compositions.

To determine which proteins are expressed in each tissue type, we performed differential expression analysis and found over 500 proteins that were differentially expressed between the samples (*Figure 2A*, *Supplementary file 8*). To identify proteins that are uniquely expressed by spindles, we compared between spindle and nerve samples and between spindle and muscle samples, and identified 387 and 40 differentially expressed proteins, respectively. To correlate these differentially expressed proteins to the different tissues of the spindle, we examined their expression in all three samples (*Figure 2B*, *Figure 2—figure supplement 1B*). Clustering of the 387 spindle-nerve differentially expressed proteins showed that the spindle sample clustered with the muscle sample, suggesting that these proteins are expressed by the muscle tissue of the spindle (*Figure 2—figure supplement 1B*).

The 40 proteins that were differentially expressed in spindle vs. muscle samples were grouped into five distinct clusters (*Figure 2B*). Cluster 1 contains proteins expressed in both spindle and nerve, but not in extrafusal fibers, suggesting that these proteins are expressed in the neuronal tissue of the spindle. The expression of the pan-neuronal marker Tubb3 (*Lee et al., 1990*) in this cluster supports this assumption. Cluster 2 contains proteins that were expressed in the spindle and had almost no expression in either muscle or nerve samples. This cluster includes extracellular proteins such as versican and elastin, suggesting that it may represent the spindle capsule. Cluster 3 mainly contains myosins, suggesting that it includes proteins expressed in the intrafusal fibers. Clusters 4 and 5 contain proteins highly expressed in the muscle sample as compared to the spindle sample. Interestingly, clusters 1 and 2 contain 24 proteins that were expressed by spindles but not by extrafusal fibers. Since spindles are sporadically embedded within the muscle, these 24 proteins may serve as spindle-specific markers.

Finally, to correlate between RNA-seq and proteomic data, we compared the lists of spindle-muscle differentially expressed genes (*Figure 1*) and proteins (*Figure 2B*). The results showed that 29 molecules were differentially expressed in the same direction at both protein and RNA levels (*Figure 2C*, red dots).

Taken together, our proteomic data identified 40 proteins that are differentially expressed between the muscle spindle and the surrounding muscle tissue. Twenty-four of these proteins were expressed only in the spindle, suggesting them as potential markers for these proprioceptors.

## Myl2, Atp1a3, VCAN, and Glut1 are new markers for different muscle spindle tissues

To identify markers for the different tissues of the spindle, we searched for differentially expressed molecules that were found to be upregulated in both our transcriptomic and proteomic analyses (*Figure 2C*). Next, we associated 16 of the 22 detected molecules to their predicted tissue by crossing them with RNA datasets of proprioceptive neurons, γ-motoneurons, extracellular genes, and intrafusal fibers (*Figure 1*, *Figure 3—figure supplement 1*). Four out of the remaining six candidates are contractile proteins, suggesting that they are expressed by intrafusal fibers. The two remaining proteins were classified as 'other' (*Figure 3A*, *Figure 3—figure supplement 1*).

To validate our predictions, we studied the expression of several potential markers using immunofluorescence staining on section of deep masseter muscle (*Figure 3—figure supplement 2*) and on whole extensor digitorum longus (EDL) muscle. Samples were subjected to a clearing protocol that allows visualization and analysis of intact muscle spindles (*Figure 3*, *Figure 3—figure supplement 2*).

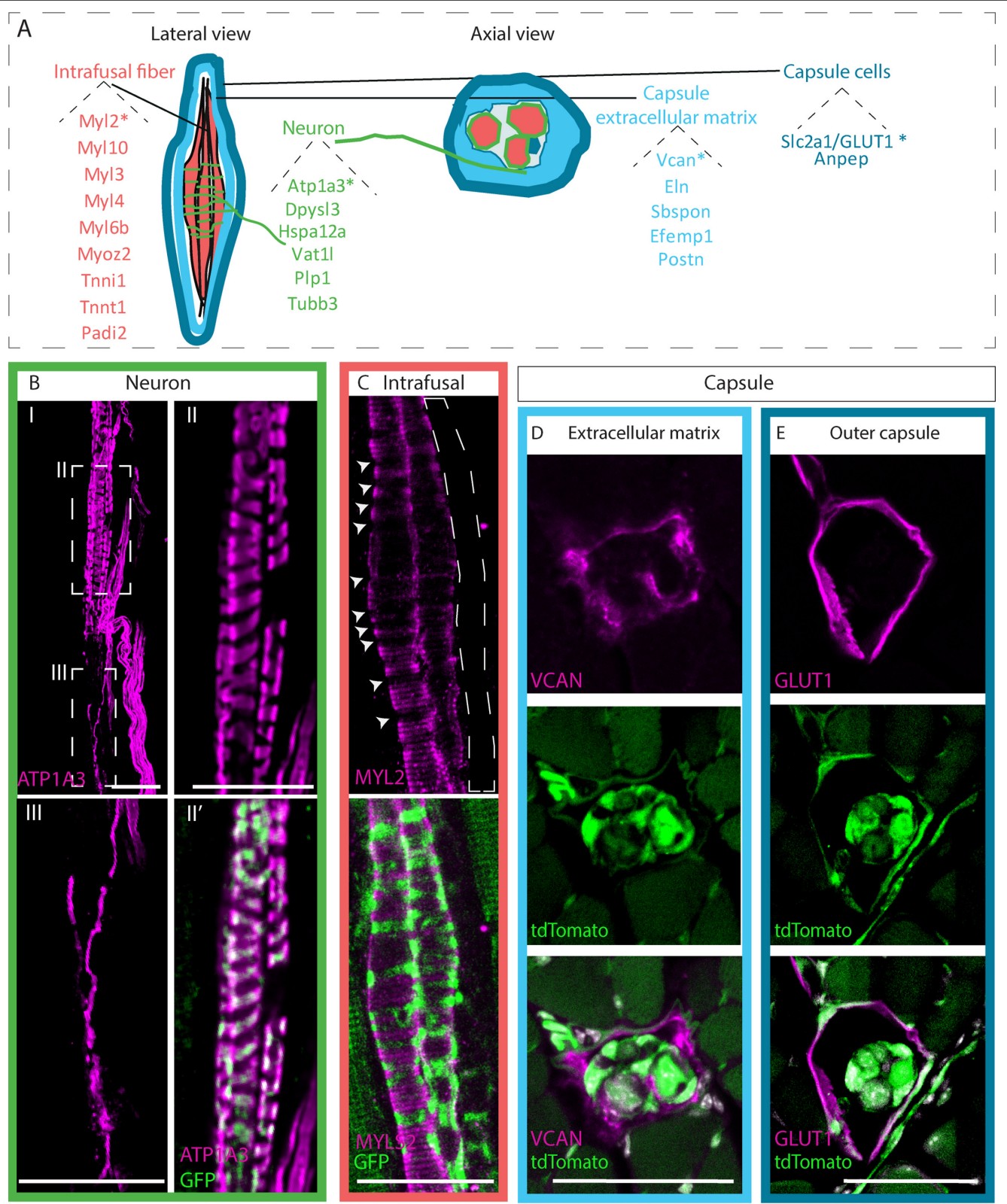

**Figure 3.** Identifying novel molecular markers for different muscle spindle tissues. (**A**) Schematic representation of the different tissues composing the muscle spindle in a lateral (left) and axial (center) views. Twenty-two potential markers that were found to be upregulated at both RNA and protein levels are listed next to their predicted tissue of expression, namely intrafusal fibers (red), neurons (green), capsule cells (dark blue), and capsule extracellular matrix (light blue). Markers that were further validated are marked with asterisks. (**B,C**) Confocal images of whole-mount extensor digitorum longus (EDL)

*Figure 3 continued on next page*

*Figure 3 continued*

muscle from *Piezo2^EGFP-IRES-Cre* mice, in which proprioceptive neurons are fluorescently labeled by GFP (green), which were immunostained for ATP1A3 (B, magenta) or myosin light chain 2 (MYL2) (C, magenta). Anti-ATP1A3 stained proprioceptive neurons (BII) and γ-motoneurons (BIII). (II,III) are high magnifications of the boxed areas in (I). Anti-MYL2 stained intrafusal bag fibers, but not chain fibers (indicated by dashed lines); arrowheads indicate the neuron-muscle interface, where MYL2 staining was absent. Scale bars represent 50 μm. (D,E) Confocal images of transverse sections of forelimb muscles from *Piezo2^EGFP-IRES-Cre*;*Rosa26^tdTomato* mice, in which muscle spindles are fluorescently labeled by tdTomato (green), which were immunostained for versican (VCAN) (D, magenta) or GLUT1 (E, magenta). VCAN was expressed in the extracellular matrix of the capsule, whereas GLUT1 expression was restricted to the outer capsule cells. Scale bars represent 50 μm.

The online version of this article includes the following figure supplement(s) for figure 3:

**Figure supplement 1.** Correlation between protein and RNA upregulation in the spindle.

**Figure supplement 2.** Validation of predicted muscle spindle markers.

**Figure supplement 3.** Examination of the expression of muscle spindle markers in the Golgi tendon organ (GTO).

First, we examined the neuronal marker ATPase, Na+/K+ transporting, alpha 3 polypeptide (ATP1A3), which was predicted by our analysis to be expressed in both proprioceptive and γ-moto-neurons (*Figure 3—figure supplement 1*). In agreement with our prediction, ATP1A3 was previously shown to be expressed in proprioceptive sensory neurons (*Romanovsky et al., 2007*) and γ-moto-neurons (*Edwards et al., 2013*). Indeed, immunostaining of deep masseter sections showed ATP1A3 expression by proprioceptive neurons in the central part of the spindle (*Figure 3—figure supplement 2A*). Whole-mount staining revealed ATP1A3 expression in both proprioceptive neurons (*Figure 3BI, II*, *Figure 3—figure supplement 2B*) and γ-motoneurons (*Figure 3BIII*). These results demonstrate our ability to identify potential marker from our multi-omic data.

As a marker for intrafusal fibers, we tested myosin light chain 2 (MYL2). We found that within the deep masseter muscle, MYL2 expression was restricted to intrafusal fibers and was excluded from the extrafusal fibers (*Figure 3—figure supplement 2C*). Whole-mount staining showed that MYL2 was expressed by all bag fibers (*Figure 3C*, *Figure 3—figure supplement 2D*) and that its levels were reduced in areas that are contacted by proprioceptive neurons (*Figure 3C*, arrowheads).

As a potential marker for capsule ECM, we studied the expression of versican (VCAN). Immu-nostaining of deep masseter sections, whole-mount EDL (*Figure 3—figure supplement 2E–F*), and transverse sections of forelimbs (*Figure 3D*) showed that VCAN is expressed in the extracellular space between the inner and outer capsule.

Next, we analyzed the expression of solute carrier family 2 member 1 (*slc2a1*), that encodes GLUT1 protein from the 'other' group. We found it to be expressed along the outer boundaries of the spindle (*Figure 3—figure supplement 2G and H*). To localize its expression more accurately, we examined transverse sections of the spindle and found that GLUT1 is expressed by outer capsule cells (*Figure 3E*).

Finally, as both muscle spindles and GTOs are found in the musculoskeletal system (*Proske and Gandevia, 2012*), we studied the expression of our validated markers in GTOs. We found that ATP1a3, VCAN, and GLUT1 were also detected in the different GTO tissues (*Figure 3—figure supplement 3*).

Taken together, these results highlight and validate new markers for the different tissues composing the muscle spindle and GTO, which may serve for imaging and studying these tissues.

## Analysis of new markers during development revealed sequential steps of spindle differentiation

To date, little is known on the molecular events that take place during muscle spindle development. Having identified new markers for spindle tissues, we proceeded to utilize these markers to study the postnatal development of these proprioceptors. At this stage, proprioceptive neurons, intrafusal fibers, and capsule cells are already present (*Milburn, 1973*). Using antibodies against these markers, we stained sections of deep masseter muscles from *Thy1-YFP* reporter mice (*Feng et al., 2000*), where neuronal tissue is fluorescently labeled, at postnatal days (P) 3, 7, and 25. Examination showed that at P3, only the neuronal marker ATP1A3 was expressed in the spindle (*Figure 4A*). Further examination showed that ATP1A3 was expressed prenatally as early as E15.5 (*Figure 4—figure supplement 1*). By P7, the capsule cell marker GLUT1 was prominently expressed in the outer capsule (*Figure 4B*) and by P25, the capsule ECM marker VCAN was expressed in the extracellular space (*Figure 4C*).

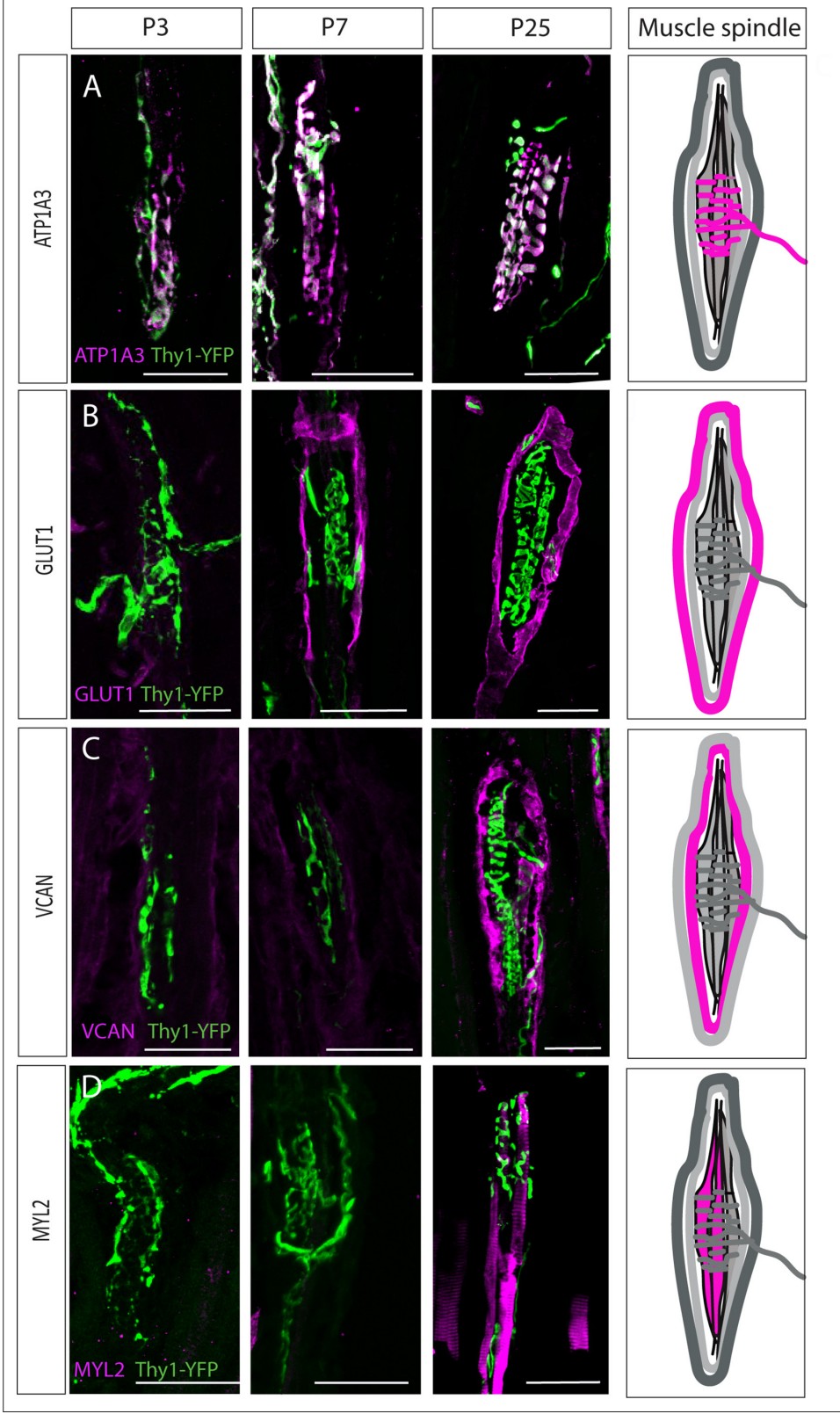

**Figure 4.** Postnatal muscle spindle development. (**A–D**) Confocal images of longitudinal sections of the deep masseter muscle from *Thy1-YFP* mice (YFP, green) stained with the indicated antibodies at P3, P7, and P25. ATP1A3 (**A**) was expressed by proprioceptive neurons at all examined time points. GLUT1 expression (**B**) was detected in the outer capsule cells at postnatal day 7 (P7). Versican (VCAN) expression (**C**) was detected in the extracellular

*Figure 4 continued on next page*

*Figure 4 continued*

space at P25. Myosin light chain 2 (MYL2) expression (**D**) was detected in intrafusal fibers at P25. ATP1A3, GLUT1, VCAN, and MYL2 are in magenta. Scale bars represent 50 μm. On the right, schematic representations of adult muscle spindle with the analyzed tissue in magenta.

The online version of this article includes the following figure supplement(s) for figure 4:

**Figure supplement 1.** Embryonic expression of ATP1a3.

**Figure supplement 2.** Sarcomeric organization of the muscle spindle.

The intrafusal marker MYL2 was also first detected at P25 (*Figure 4D*). To determine if the sarcomeric organization of the intrafusal fibers forms prior to P25, we stained P3, P7, and P25 intrafusal fibers with phalloidin. As seen in *Figure 4—figure supplement 2*, this organization was observed already at P3, suggesting that the identity of the intrafusal fibers is acquired after the myofiber has been established. Taken together, these results suggest that during spindle development, capsule cells and intrafusal fibers undergo several differentiation steps to reach their mature state.

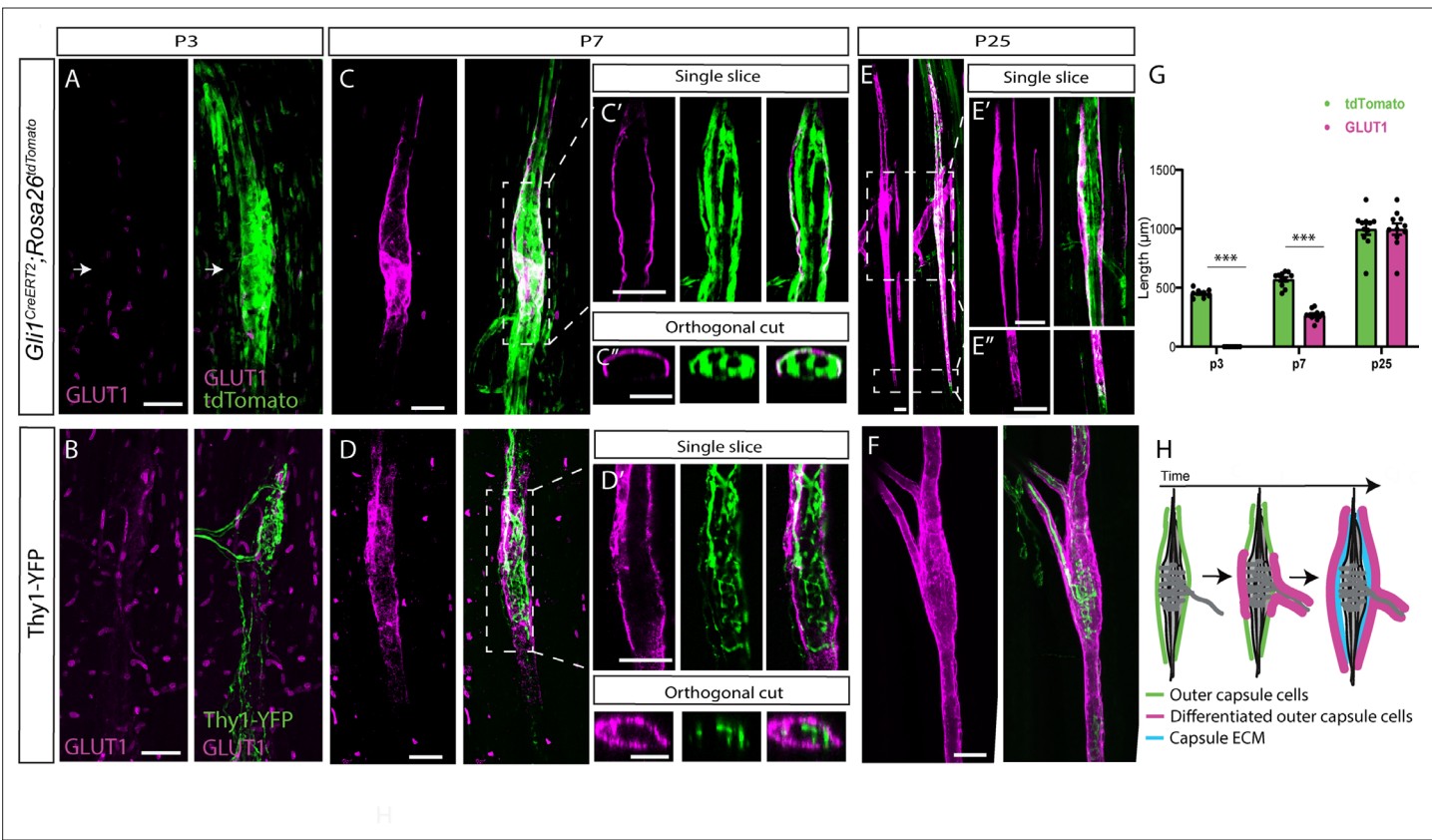

**Figure 5.** Postnatal development of muscle spindle capsule cells. (**A,C,E**) Confocal images of whole-mount extensor digitorum longus (EDL) muscle taken from *Gli1^CreERT2;Rosa26^tdTomato* mice stained for GLUT1 (magenta). *Gli1⁺* were labeled by a single tamoxifen administration at postnatal day 1 (P1) and analyzed at P3 (**A**), P7, (**C**) and P25 (**E**). Arrows in (**A**) show the location of the capsule cells, as marked by TdTomato (green). The center of the spindle (boxed area in C) is shown in a single slice (**C'**) and in an orthogonal view (**C"**). The center of the spindle (top boxed area in E) and its edges (bottom boxed area in E) are shown in single slices (E' and E", respectively). (**B,D,F**) Confocal images of whole-mount EDL muscle from *Thy1-YFP* mice stained with anti-GLUT1 antibody at P3 (**B**), P7 (**D**), and P25 (**F**). The center of the spindle (boxed area in D) is shown in a single slice (**D'**) and in an orthogonal view (**D"**). (**G**) Quantification of capsule length, as measured in (**A,C,E**) based on GLUT1 and tdTomato labeling ($n_{P3}$ = 7, p = 9.07E-08; $n_{P7}$ = 10, p = 1.09E-08; $n_{P25}$ = 10, p = 0.989; two-tailed *t*-test; data are presented as mean ± SEM; each dot represents one spindle). (**H**) Schematic representation of the maturation process of the outer capsule throughout postnatal development. Scale bars represent 50 μm in (**A–F**) and 20 μm in (**B',C'**).

## Differentiation of muscle spindle capsule cells propagates from the center toward the polar ends

Next, we studied the development of the spindle capsule. It was previously suggested that the outer spindle capsule is part of the perineural sheath in the peripheral nervous system (*Shantha et al., 1968*). Recently, *Gli1* was shown to be expressed by perineural glial cells (*Zotter et al., 2022*). Examination of our data showed that *Gli1* transcripts were significantly upregulated in the muscle spindle samples as compared to the extrafusal fiber samples (log fold change 3.57; *Supplementary file 1*). To determine whether *Gli1* is expressed by capsule cells, we crossed *Gli1^{CreERT2}* mice (*Ahn and Joyner, 2004*) with *Rosa26^{tdTomato}* reporter (*Madisen et al., 2010*). We administrated tamoxifen to *Gli1^{CreERT2};Rosa26^{tdTomato}* mice at P1 and analyzed tdTomato expression at P3 in whole-mount preparations of the EDL muscle. As seen in *Figure 5A*, tdTomato was extensively expressed around the spindle, suggesting that at P1, *Gli1*-positive cells contribute to the developing capsule.

Next, we analyzed spatially the differentiation sequences of capsule cells by staining whole-mount EDL muscles for GLUT1. We used either *Gli1^{CreERT2};Rosa26^{tdTomato}* mice that were administered tamoxifen at P1, or *Thy1-YFP* mice. At P3, we observed TdTomato expression by capsule cells and YFP expression in proprioceptive neurons at the center of the spindle. However, in agreement with our results in the masseter (*Figure 4*), we did not detect GLUT1 expression around the spindle (*Figure 5A and B*). At P7, while TdTomato expression extended to the spindle periphery (*Figure 5C*, quantified in G), GLUT1 expression was restricted to the central domain of the spindle, covering the coil structure formed by proprioceptive neurons (*Figure 5D*). Examination of single optical sections of the capsule showed that GLUT1 was co-expressed with tdTomato, suggesting that *Gli1*-positive capsule cells also express *Glut1* (*Figure 5C'*). By P25, GLUT1 expression extended from the center and covered the polar ends of the spindle in a similar pattern to the tdTomato-positive cells (*Figure 5E'*, quantified in *Figure 5G*).

Taken together, these results reveal the spatiotemporal sequence of the differentiation of *Gli1+* during muscle spindle capsule development and maturation (*Figure 5H*).

## Discussion

Although the muscle spindle is the main sensory organ of proprioception, its molecular composition has only recently started to be unraveled. Here, we generated comprehensive transcriptomic and proteomic datasets of the entire muscle spindle. By analyzing these datasets, we have identified a set of new markers for the different tissues that compose the spindle. Finally, using these markers, we studied the development of the capsule that envelops the spindle and revealed several differentiation steps. Together, these findings provide a new tool for future studies of muscle spindle biology and pathology.

For many years, the molecular composition of muscle spindle was neglected. In recent years, the molecular identities of proprioceptive sensory neurons, γ-motoneurons, and intrafusal fibers have been uncovered (*Blum et al., 2021*; *Kim et al., 2020*; *Oliver et al., 2021*; *Wu et al., 2019*; *Wu et al., 2021*). Although these studies advanced our understanding of these tissues, a comprehensive, organ-level account of gene expression profiles of all tissues composing the spindle was still missing. Several factors contributed to this shortage. First, there was no transcriptional data on the muscle spindle capsule cells. Second, proprioceptive sensory neurons and γ-motoneurons RNA data were collected from their soma, not necessarily reflecting the expression in the neuron endings. Finally, most of these studies were performed at a single-cell or single-nucleus resolution, hence lacking deep RNA coverage. In this study, we filled these gaps by providing deep sequencing data combined with proteomic data of the intact muscle spindle.

Local translation of mRNA into protein was shown to regulate multiple aspects of axonal and dendritic biology (*Holt and Schuman, 2013*). Because our study was performed on intact muscle spindle, including neuron endings, we could identify genes and proteins that are most likely expressed in terminals. By comparing our RNA-seq data to the expression profiles of isolated DRG proprioceptive neurons (*Zheng et al., 2019*) and γ-motoneurons (*Blum et al., 2021*), we identify RNA transcripts that potentially derive from these two neuron types. Interestingly, GO enrichment analysis identified a number of enriched synapse-related terms in the neuronal endings of both proprioceptive neurons and γ-motoneurons. Indeed, the peripheral endings of proprioceptive neurons were shown to have

presynapse-like structures (*Bewick et al., 2005*; *Bewick and Banks, 2014*; *Than et al., 2021*; *Zhang et al., 2014*). The presence of many presynaptic genes in our RNA dataset reinforces these findings and provides a strong indication to our ability to collect data from these terminals.

The main problem in the study of muscle spindle is the lack of molecular markers for its different compartments. Our study provides a list of putative markers for the different spindle tissues, which was generated by crossing between our transcriptomic and proteomic datasets. Several of these candidates, which were verified using immunohistochemistry, can be used to study spindle development and pathology.

The biology of the capsule and the cells that compose it is largely unknown. Questions relating, for example, to the origin of capsule cells, their level of heterogeneity, or the coordination of their differentiation with the development of other spindle tissues are still open. Indication of the significance of the capsule comes from studies on muscular dystrophy, where capsule thickening is observed in both patients (*Cazzato and Walton, 1968*; *Kararizou et al., 2007*) and mice (*Ovalle and Dow, 1986*). By using the markers we have identified, we provide spatiotemporal information on capsule development. We show that the outer capsule cells undergo several steps of differentiation and ECM secretion. Moreover, we found that spatially, the differentiation wave starts at the center of the spindle and propagates to its polar ends. These finding are in agreement with previous studies showing that intrafusal fibers and neurons undergo postnatal maturation (*Maeda et al., 1985*; *Soukup et al., 1995*) and indicate developmental coordination between the different tissues of the spindle. It is tempting to speculate about the nature of the molecular mechanism that coordinates this process, which we predict to involve secreted molecules. Interestingly, several secreted growth factors, such as BMPs and FGFs, were identified by our transcriptomic analysis and are therefore potential candidates to coordinate spindle development.

The spindle capsule is composed of ECM molecules such as collagens type IV and VI (*Maier and Mayne, 1987*; *Ovalle and Dow, 1985*; *Sanes, 1982*). Our database contains many ECM genes, some of which likely contribute to the capsule. For example, we showed that VCAN is expressed in the capsule ECM. Furthermore, we identified two novel capsule cell markers, namely GLUT1 for outer capsule cells and Gli1 for outer and inner capsule cells. These findings raise questions regarding the origin of the capsule cells. Since muscle spindles contain both muscle and neuronal tissues, capsule cells may originate either in muscle mesenchyme or in the nervous system. Moreover, the difference we observed between gene expression profiles in inner and outer capsule raise the possibility that these are two distinct cell populations of different origins. Indeed, several lines of evidence suggest that capsule cells have two different origins within the nervous system. First, ultrastructural studies using electron microscopy identified high similarity between outer capsule and perineurial cells and between inner capsule cells and endoneurial fibroblasts of the peripheral nervous system (*Dow et al., 1980*; *Edwards, 1975*). Indeed, we show that the outer capsule cells express perineurial markers, further supporting the perineurial origin of the outer capsule. However, to verify this hypothesis and rule out muscle origin, detailed lineage tracing studies are needed.

Recent findings show that the proprioceptive system and, specifically, muscle spindles play vital regulatory roles in skeletal development and function (*Assaraf et al., 2020*; *Blecher et al., 2017a*; *Blecher et al., 2017b*). These findings suggest that spindle pathology would have a broad effect on the musculoskeletal system, highlighting the importance of uncovering the molecular composition of the muscle spindle. The molecular data we provide herein will thus support future studies of muscle spindle as well as musculoskeletal biology.

## Materials and methods

**Key resources table**

| Reagent type (species) or resource | Designation | Source or reference | Identifiers | Additional information |
|---|---|---|---|---|
| Gene (*Mus musculus*) | B6(SJL)-Piezo2$^{tm1.1(cre)Apat}$/J | Jackson Laboratory | Stock #027719 RRID:IMSR_JAX:027719 | |
| Gene (*Mus musculus*) | Gli1$^{tm3(cre/ERT2)Alj}$/J | Jackson Laboratory | Strain #:007913 RRID:IMSR_JAX:007913 | |

*Continued on next page*

*Continued*

| Reagent type (species) or resource | Designation | Source or reference | Identifiers | Additional information |
|---|---|---|---|---|
| Gene (*Mus musculus*) | B6.Cg-Gt(ROSA)26Sor^tm9(CAG-tdTomato)Hze^/J | Jackson Laboratory | Strain #:007909 RRID:IMSR_JAX:007909 | |
| Gene (*Mus musculus*) | B6.Cg-Tg(Thy1-YFP)16Jrs/J | Jackson Laboratory | Strain #:003709 RRID:IMSR_JAX:003709 | |
| Antibody | Anti-ATP1a3 (rabbit polyclonal) | Millipore | Cat# 06-172I, RRID:AB_310066 | Section 1:100 Whole mount 1:100 |
| Antibody | Anti-VERSICAN (rabbit polyclonal) | Abcam | Cat# ab19345, RRID:AB_444865 | Section 1:300 Whole mount 1:50 |
| Antibody | Anti-MYL2 (rabbit polyclonal) | Abcam | Cat# ab79935, RRID:AB_1952220 | Section 1:100 Whole mount 1:100 |
| Antibody | Anti-GLUT1 (rabbit monoclonal) | Abcam | Cat# ab195020, RRID:AB_2783877 | Section 1:400 Whole mount 1:200 |
| Antibody | Anti-GFP (biotin goat polyclonal) | Abcam | Cat# ab6658 RRID:AB_305631 | Section 1:100 Whole mount 1:100 |
| Antibody | Cy5 conjugated donkey anti-rabbit (polyclonal) | Jackson ImmunoResearch Laboratories | Cat# 711-175-152 RRID:AB_2340607 | Section 1:100 Whole mount 1:200 |
| Peptide, recombinant protein | Phalloidin, synthetic peptide (TRITC) | Sigma-Aldrich | Cat# P1951 RRID:AB_2315148 | Section 2 μg/ml |
| Peptide, recombinant protein | Native Streptavidin protein (DyLight 488) | Abcam | Cat# ab134349 | Section 1:100 Whole mount 1:200 |
| Software, algorithm | ImageJ software | ImageJ (http://imagej.nih.gov/ij/) | RRID:SCR_003070 | |

## Mouse lines

All experiments involving mice were approved by the Institutional Animal Care and Use Committee (IACUC) of the Weizmann Institute. Mice were housed in a temperature- and humidity-controlled vivarium on a 12 hr light-dark cycle with free access to food and water.

The following strains were used: *Piezo2^EGFP-IRES-Cre^* (The Jackson Laboratory, #027719), *Gli1^CreERT2^* (The Jackson Laboratory, #007913), *Rosa26^tdTomato^* (The Jackson Laboratory, #007909), and *Thy1-YFP16* (The Jackson Laboratory, #003709).

In all experiments, at least three mice from different litters were used. Mice were genotyped by PCR of genomic DNA from ear clips. Primer sequences and amplicon sizes are listed in **Table 1**.

## Muscle spindle isolation

To isolate entire muscle spindles, spindles were labeled using the *Piezo2^EGFP-IRES-Cre^* reporter driving the expression of *tdTomato*. Mice were sacrificed and deep masseter muscle was manually exposed and dissected in ice-cold Liley's solution (**Liley, 1956**) (NaHCO$_3$ 1 g, KCl 0.3 g, KH$_2$PO$_4$ 0.13 g, NaCl 0.2 g, CaCl$_2$ 1 M 2 ml, all adjusted to 1 l DDW) on a silicone-coated plate (Sylgard 184 silicone elastomer base). Muscle spindle bundles, each containing about 20 spindles, were microdissected under fluorescent microscope.

## Proteomic analysis

For proteomic analysis, samples of muscle spindles (fluorescently labeled), adjacent extrafusal muscle (non-florescent), and the nerve bundle innervating the muscle (fluorescently labeled) were isolated. Collected samples were immediately frozen in liquid nitrogen. For each tissue type, six samples were collected from one deep masseter muscle of six different mice.

## Sample preparation

Samples for protein profiling were prepared at the Crown Genomics Institute of the Nancy and Stephen Grand Israel National Center for Personalized Medicine, Weizmann Institute of Science.

**Table 1.** Primer sequences and amplicon sizes used for PCR.

| Reaction | Amplicon (bp) | Sequences |
|---|---|---|
| Cre | 800 | F: CCTGGAAAATGCTTCTGTCCGTTTGCC<br>R::GAGTTGATAGCTGGCTGGTGGCAGATG |
| Cre-ER$^{T2}$ | 800 | F: CCTGGAAAATGCTTCTGTCCGTTTGCC<br>R: GAGTTGATAGCTGGCTGGTGGCAGATG |
| tdTomato (wild type) | 297 | F: AAG GGA GCT GCA GTG GAG TA<br>R: CCG AAA ATC TGT GGG AAG TC |
| tdTomato (tdTomato-flox allele) | 196 | F: GGC ATT AAA GCA GCG TAT CC<br>R: CTG TTC CTG TAC GGC ATG G |
| YFP (GFP) | 300 | F: GACGGCAACATCCTGGGGCACAAG<br>R: CGGCGGCGGTCACGAACTCC |

Samples were subjected to in-solution tryptic digestion using the suspension trapping (S-trap) as previously described (*Elinger et al., 2019*). Briefly, tissue was homogenized in the presence of lysis buffer containing 5% SDS in 50 mM Tris-HCl. Lysates were incubated at 96°C for 5 min, followed by six cycles of 30 s of sonication (Bioruptor Pico, Diagenode, USA). Protein concentration was measured using the BCA (Thermo Fisher Scientific, USA). Then, 50 µg of total protein was reduced with 5 mM dithiothreitol and alkylated with 10 mM iodoacetamide in the dark. Each sample was loaded onto S-trap microcolumns (Protifi, USA) according to the manufacturer's instructions. After loading, samples were washed with 90:10% methanol/50 mM ammonium bicarbonate. Samples were then digested with trypsin (1:50 trypsin/protein) for 1.5 hr at 47°C. The digested peptides were eluted using 50 mM ammonium bicarbonate. Trypsin was added to this fraction and incubated overnight at 37°C. Two more elutions were made using 0.2% formic acid and 0.2% formic acid in 50% acetonitrile. The three elutions were pooled together and dried by vacuum centrifugation. Samples were kept at −80°C until further analysis.

## Liquid chromatography

ULC/MS grade solvents were used for all chromatographic steps. Dry digested samples were dissolved in 97:3% $H_2O$/acetonitrile containing 0.1% formic acid. Each sample was loaded using split-less nano-ultra performance liquid chromatography (nanoUPLC; 10 kpsi nanoAcquity; Waters, Milford, MA, USA). The mobile phase was: (A) $H_2O$ with 0.1% formic acid and (B) acetonitrile with 0.1% formic acid. Samples were desalted online using a reversed-phase Symmetry C18 trapping column (180 µm internal diameter, 20 mm length, 5 µm particle size; Waters). The peptides were then separated using a T3 HSS nano-column (75 µm internal diameter, 250 mm length, 1.8 µm particle size; Waters) at 0.35 µl/min. Peptides were eluted from the column into the mass spectrometer using the following gradient: 4–20% B in 155 min, 20–90% B in 5 min, maintained at 90% for 5 min and then back to initial conditions.

## Mass spectrometry

The nanoUPLC was coupled online through a nanoESI emitter (10 µm tip; New Objective; Woburn, MA, USA) to a quadrupole orbitrap mass spectrometer (Q Exactive HFX, Thermo Fisher Scientific) using a FlexIon nanospray apparatus (Proxeon). Data were acquired in data-dependent acquisition mode, using a top 10 method. MS1 resolution was set to 120,000 (at 200 m/z), mass range of 375–1650 m/z, AGC of 3e6, and maximum injection time was set to 60 ms. MS2 resolution was set to 15,000, quadrupole isolation 1.7 m/z, AGC of 1e5, dynamic exclusion of 40 s, and maximum injection time of 60 ms.

## Data processing

Raw data were processed with MaxQuant v1.6.0.16 (*Cox and Mann, 2008*). The data were searched with the Andromeda search engine against the mouse (*Mus musculus*) protein database as downloaded from Uniprot (https://www.afternic.com/forsale/uniprot.com?utm_source=TDFS&utm_medium=sn_affiliate_click&utm_campaign=TDFS_Affiliate_namefind_direct8&traffic_type=CL3&traffic_id=Namefind), and appended with common lab protein contaminants. Enzyme specificity was set to trypsin and up to two missed cleavages were allowed. Fixed modification was set to carbamidomethylation of

cysteines and variable modifications were set to oxidation of methionines, and deamidation of gluta-mines and asparagines. Peptide precursor ions were searched with a maximum mass deviation of 4.5 ppm and fragment ions with a maximum mass deviation of 20 ppm. Peptide and protein identifications were filtered at an FDR of 1% using the decoy database strategy (MaxQuant's 'Revert' module). The minimal peptide length was seven amino acids and the minimum Andromeda score for modified peptides was 40. Peptide identifications were propagated across samples using the match-between-runs option checked. Searches were performed with the label-free quantification option selected. Decoy hits were filtered out.

## Analysis of proteomic data

Bioinformatic analysis of the proteomic data was applied on LFQ intensities of 1100 proteins, detected from all samples. Proteins with at least one razor and unique peptides were considered, removing known contaminants and reversed entries, one outlier sample was excluded from anal-ysis based on PCA. To detect differential proteins, ANOVA test was applied on log2-transformed intensities, following a multiple test correction (FDR step-up) using Partek Genomics Suite 7.0. For each pairwise comparison, we considered proteins with at least three valid measurements (out of five) in both groups that passed the thresholds of |linear fold change|≥2 and FDR ≤0.05. In addi-tion, proteins that were detected in at least three replicates in one group and completely absent in the other group were also considered as qualitatively differential proteins. For visualization of protein expression, heatmaps were generated using Partek Genomics Suite, with log2-transformed LFQ intensities, applying row standardization and partition clustering using the k-means algorithm (Euclidean method). Scatter plots between the fold change of protein and genes were calculated using the imputed protein values.

## Bulk RNA-seq

For RNA analysis, muscle spindle bundles and adjacent extrafusal muscle fibers were collected and immediately frozen in liquid nitrogen. From each tissue type, six samples from six mice were produced.

## Sample preparation

Total RNA was purified using Qiazol followed by chloroform phase separation and application of RNeasy Micro kit (Qiagen). RNA quality and concentration was measured by NanoDrop and TapeS-tation. RNA-seq libraries were prepared at the Crown Genomics Institute of the Nancy and Stephen Grand Israel National Center for Personalized Medicine, Weizmann Institute of Science. Libraries were prepared using the INCPM-mRNA-seq protocol. Briefly, the polyA fraction (mRNA) was purified from 500 ng of total input RNA followed by fragmentation and generation of double-stranded cDNA. After Agencourt Ampure XP beads cleanup (Beckman Coulter), end repair, A base addition, adapter ligation, and PCR amplification steps were performed. Libraries were quantified by Qubit (Thermo Fisher Scientific) and TapeStation (Agilent). Sequencing was done on a NextSeq instrument (Illumina) using a single end 84 cycles high output kit, allocating ~20 M reads or more per sample (single read sequencing).

## Analysis of RNA-seq data

Transcriptomic data from five replicates of spindle samples and six muscle samples were analyzed. One spindle sample was omitted from the analysis because of low number of reads uniquely aligned to genes. A user-friendly Transcriptome Analysis Pipeline (UTAP) version 1.10.1 was used for analysis (*Kohen et al., 2019*). Reads were mapped to the *M. musculus* genome (Genome Reference Consor-tium Mouse Build 38 [GRCm38], version M25 Ensembl 100) using STAR (v2.4.2a) (*Dobin et al., 2013*) and GENECODE annotation. Only reads with unique mapping were considered for further analysis. Gene expression was calculated and normalized using DESeq2 version 1.16.1 (*Love et al., 2014*), using only genes with a minimum of five reads in at least one sample. Raw p-values were adjusted for multiple testing (*Benjamini and Hochberg, 1995*). A gene was considered differentially expressed if it passed the following thresholds: minimum mean normalized expression of 5, adjusted p-value ≤0.05, and absolute value of log2 fold change ≥1.

## Ingenuity pathway analysis

Genes were classified according to cell location using QIAGEN Ingenuity Pathway Analysis algorithm (https://digitalinsights.qiagen.com/; Qiagen, Redwood City, CA; USA; *Krämer et al., 2014*).

## Gene set enrichment analysis

To identify differentially expressed upregulated spindle genes that were enriched in DRG proprioceptive neurons or γ-motoneurons, pre-ranked GSEA (*Subramanian et al., 2005*) was performed using default settings against either DRG proprioceptive neurons genes (total of 37,729 genes; *Zheng et al., 2019*) or γ-motoneurons genes (total of 21,455 genes; *Blum et al., 2021*) sorted by expression values.

## Gene ontology

Go enrichment analysis was done using Metascape web tool (https://metascape.org/) choosing GO terms with p-value <0.05.

## Immunofluorescence of cryosections

For immunofluorescence, mice were sacrificed and fixed overnight in 4% paraformaldehyde (PFA)/PBS at 4°C. For longitudinal cryosectioning, the deep masseter muscle was dissected, transferred to 30% sucrose overnight, then embedded in OCT and sectioned by cryostat at a thickness of 10–20 μm. For transverse cryosection immunofluorescence, forelimbs were dissected, incubated with 0.5 mol/l EDTA (pH 7.4) for 2 weeks for decalcification, transferred to 30% sucrose overnight, then embedded in OCT by orienting the humerus at 90° to the plate, and sectioned by cryostat at a thickness of 10 μm.

Cryosections were dried and post-fixed for 10 min in 4% PFA, permeabilized with PBS with 0.3% Triton X-100, washed with PBS with 0.1% Tween-20 (PBST) for 5 min and blocked with 7% goat/horse serum and 1% bovine serum albumin (BSA) dissolved in PBST. Then, sections were incubated with primary antibody (see Key resources table) at 4°C overnight. The next day, sections were washed three times in PBST and incubated for 1 hr with secondary antibody conjugated fluorescent antibody, washed three times in PBST, counterstained with DAPI, and mounted with Immu-mount aqueous-based mounting medium (Thermo Fisher Scientific).

## Whole-mount immunofluorescence

For whole-mount immunofluorescence, muscles were subjected to optical tissue clearing protocol for mouse skeletal muscle adapted from *Williams et al., 2019*. Briefly, post-fixed EDL muscle was dissected, washed in PBS, and placed in an A4P0 hydrogel (4% acrylamide, 0.25% 2'-azobis[2-(2-imidazolin-2-yl)propane]dihydrochloride in PBS) shaking at 4°C overnight. Then, hydrogel was allowed to polymerize for 3 hr at 37°C. After polymerization, the samples were washed in PBS, transferred to 5 ml of 10% SDS (pH 8.0) with 0.01% sodium azide, and were shaken gently at 37°C for 3 days to remove lipid.

Cleared samples were washed with wash buffer (PBS containing 0.5% Tween-20) for 20 min, permeabilized with PBST (PBS containing 0.3% Triton X-100) for 20 min and washed again with wash buffer for 20 min, all at room temperature shaking. Then, samples were blocked with 6% BSA dissolved in PBS containing 0.3% Triton X-100 and 0.5% Tween-20 for 2 days at 37°C shaking gently. Samples were subjected to primary antibodies (see Key resources table) for 5 days at 37°C, shaking gently, washed with wash buffer for 2 days at room temperature with frequent solution changes, incubated with secondary antibodies and DAPI for 5 days at 37°C, shaking gently, and washed again with wash buffer for 2 days at room temperature with frequent solution changes. For clearing and mounting, the samples were then incubated in 500 μl refractive index matching solution (RIMS; 74% wt/vol Histodenz in 0.02 M phosphate buffer) for 1 day at room temperature, shaking gently. Samples were mounted in RIMS and imaged using Zeiss LSM800 or LSM900 confocal microscope. Images were processed with ImageJ 1.51 (National Institute of Health).

## Cell lineage analysis

Tamoxifen (Sigma-Aldrich, T-5648) was dissolved in corn oil (Sigma-Aldrich, C-8267) at a final concentration of 50 mg/ml. Cre-mediated recombination was induced at the indicated time points by administration of 125 mg/kg of tamoxifen by oral gavage (Fine Science Tools).

## Quantification of capsule size

Capsule length was measured on confocal Z-stack projections of at least 10 different spindles taken from three different mice using ImageJ 1.51. Differences in length were assessed by two-tailed *t*-test and statistical significance was defined as a p-value lower than 0.05.

## Acknowledgements

We thank Nitzan Konstantin for expert editorial assistance, Dr Alon Savidor from The Nancy and Stephen Grand Israel National Center for Personalized Medicine for his help in proteomic analysis, Drs Aaron D Gitler and Jacob A Blum from the Department of Genetics, Stanford University School of Medicine, for providing us the γ-motoneuron expression datasets, and Dr Carmen Birchmeier from the Max-Delbrück-Centrum for providing us the intrafusal expression datasets. This study was supported by grants from The David and Fela Shapell Family Center for Genetic Disorders Research, the Julie and Eric Borman Family Research Funds, and by the Nella and Leon Benoziyo Center for Neurological Diseases at the Weizmann Institute of Science.

## Additional information

### Funding

| Funder | Grant reference number | Author |
| --- | --- | --- |
| The David and Fela Shapell Family Center for Genetic Disorders Research | | Elazar Zelzer |
| The Julie and Eric Borman Family Research Funds | | Elazar Zelzer |
| The Nella and Leon Benoziyo Center for Neurological Diseases | | Elazar Zelzer |

The funders had no role in study design, data collection and interpretation, or the decision to submit the work for publication.

### Author contributions

Bavat Bornstein, Data curation, Validation, Investigation, Visualization, Writing - original draft; Lia Heinemann-Yerushalmi, Conceptualization, Data curation, Investigation; Sharon Krief, Data curation, Investigation; Ruth Adler, Investigation; Bareket Dassa, Dena Leshkowitz, Formal analysis; Minchul Kim, Resources; Guy Bewick, Robert W Banks, Methodology, Writing – review and editing; Elazar Zelzer, Conceptualization, Project administration

### Author ORCIDs

Bavat Bornstein (ID) http://orcid.org/0000-0003-0838-4603
Robert W Banks (ID) http://orcid.org/0000-0003-1614-6488
Elazar Zelzer (ID) http://orcid.org/0000-0002-1584-6602

### Ethics

All experiments involving mice were approved by the Institutional Animal Care and Use Committee (IACUC) of the Weizmann Institute (#02180222-2).

### Decision letter and Author response

Decision letter https://doi.org/10.7554/eLife.81843.sa1
Author response https://doi.org/10.7554/eLife.81843.sa2

## Additional files

### Supplementary files

• Supplementary file 1. Differentially expressed genes between spindle and muscle samples. DE,

differentially expressed.

• Supplementary file 2. Gene ontology (GO) enrichment analysis for upregulated genes in the various cellular compartment using Metascape.

• Supplementary file 3. Differentially upregulated genes localized to the extracellular space. Ingenuity pathway analysis revealed 325 differentially upregulated genes localized to the extracellular space.

• Supplementary file 4. Gene set enrichment analysis (GSEA) results. Genes of proprioceptive neurons (*Zheng et al., 2019*) and of γ-motoneurons (*Blum et al., 2021*) that contributed to the enrichment score are listed, as are 178 common genes between proprioceptive neurons and γ-motoneurons, muscle spindle-specific genes, γ-motoneuron-specific genes, and common genes between proprioceptive neurons, γ-motoneurons, and α-motoneurons.

• Supplementary file 5. Gene ontology (GO) enrichment analysis of neuronal genes (listed in *Supplementary file 4*) using Metascape.

• Supplementary file 6. Intrafusal genes. Comparison of differentially upregulated genes in our dataset with the expression profile of intrafusal fibers (*Kim et al., 2020*) revealed 187 overlapping genes.

• Supplementary file 7. Gene ontology (GO) enrichment analysis of intrafusal genes (listed in *Supplementary file 6*) using Metascape.

• Supplementary file 8. Proteomic analysis results – differentially expressed proteins between spindle, muscle, and nerve samples. The expression intensity of all detected proteins as well as differentially expressed (DE) proteins between spindle and muscle samples and between spindle and nerve samples are shown.

• MDAR checklist

### Data availability

Sequencing data have been deposited in GEO under accession number GSE208147. The raw data of proteomic profiling were deposited in the ProteomeXchange via the Proteomic Identification Database (PRIDE partner repository).

The following dataset was generated:

| Author(s) | Year | Dataset title | Dataset URL | Database and Identifier |
|---|---|---|---|---|
| Zelzer E | 2023 | Molecular characterization of the intact mouse muscle spindle using a multi-omics approach | https://www.ncbi.nlm.nih.gov/geo/query/acc.cgi?acc=GSE208147 | NCBI Gene Expression Omnibus, GSE208147 |

The following previously published datasets were used:

| Author(s) | Year | Dataset title | Dataset URL | Database and Identifier |
|---|---|---|---|---|
| Blum JA, Klemm S, Shadrach JL, Guttenplan KA, Nakayama L, Kathiria A, Hoang PT, Gautier O, Kaltschmidt JA, Greenleaf WJ, Gitler AD | 2021 | Single-cell transcriptomic analysis of the adult mouse spinal cord reveals molecular diversity of autonomic and skeletal motor neurons | https://www.ncbi.nlm.nih.gov/geo/query/acc.cgi?acc=GSE161621 | NCBI Gene Expression Omnibus, GSE161621 |
| Zheng Y, Liu P, Bai L, Trimmer JS, Bean BP, Ginty DD | 2019 | Deep Sequencing of Somatosensory Neurons Reveals Molecular Determinants of Intrinsic Physiological Properties | https://www.ncbi.nlm.nih.gov/geo/query/acc.cgi?acc=GSE131230 | NCBI Gene Expression Omnibus, GSE131230 |

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
