## [Editor Report]

This works provides a valuable and comprehensive description of the molecular composition of the different compartments of the muscle spindle. The authors combine convincing transcriptomic, proteomic and imaging approaches to provide the field with new tools for dissecting the development and function of the muscle spindle. This manuscript is of interest for a broad spectrum of researchers working on the nervous and muscular systems.

---

## [Decision Letter]

**Decision letter after peer review:**

Thank you for submitting your article "Molecular characterization of the intact muscle spindle using a multi-omics approach" for consideration by *eLife*. Your article has been reviewed by 2 peer reviewers, and the evaluation has been overseen by a Reviewing Editor Benjamin Prosser and Anna Akhmanova as the Senior Editor. The reviewers have opted to remain anonymous.

Essential revisions:

All on the reviewing team found the paper to be of high significance and quality, and with the potential to significantly benefit the field. We would be happy to consider a revised manuscript that addresses the issues listed below, with particular emphasis on the two main concerns noted by reviewer 2, which may require some additional immunofluorescence characterization.

*Reviewer #1 (Recommendations for the authors):*

– Please remember to add page and line numbers in your future manuscripts, it makes life much easier!

– In the introduction, third paragraph: the recent work of Lallemend and colleagues (Wu et al., Nat. Comm. 2021) needs to be cited along with "(Oliver et al., 2021; Wu et al., 2019)".

– Please add in the manuscript how many spindles were collected (and from how many mice) for transcriptomics and proteomics analysis.

– Results section, second paragraph: markers for intrafusal fibres, proprioceptive-SN and extrecellular matrix are discussed but not for γ-MN. It would be great to add a sentence here about the most relevant that popped out in the screen.

– "The 40 proteins that were differentially expressed in spindle vs muscle samples were grouped into five distinct clusters (Figure 2B)." In Figure 2B it says that "missing data are in white" not sure how to interpret it and there are many white boxes. Do the white boxes mean that the protein is not detected in the sample? or that there is something wrong with analysis? either way, how can you be sure this is a biological difference or just a technical issue and therefore validate the comparison? This point should be made clear.

– Figure 3BIII. How do you know apart from polar position that those fibers are γ-MN? it would be great to add a marker, VAChT?

*Reviewer #2 (Recommendations for the authors):*

Overall, the data presented and the conclusions are very significant and based on well-conducted experiments. If the authors have the space, it would also be relevant to mention the major findings in the literature regarding the establishment of the muscle spindle by myogenic regulatory factors (MRFs of the MyoD family).

Do extrafusal and intrafusal formation rely on the same mechanisms with PAX7+ myogenic stem cells enabling postnatal muscle spindle growth?

Are myotendinous junctions also required at the ends of the muscle spindles to anchor these spindles in the tendons?

Main concerns

– The authors in Figure 4 show that Myl2 is not detected: is a sarcomeric organization already established in the muscle spindle at this stage of development? This could be assessed by phalloidin staining for example and give us information on the potential biphasic specialization of this structure: a first common stage with extrafusal myofibers, then a second phase where the identity of intrafusal myofibers would be activated.

– In Figure S5, the ends of the fibers have been cut off. If the authors have pictures of these ends, they should provide them: what do the myotendinous junctions of the muscle spindles look like? In Figure 5, the ends of the myofibers are not shown, why? In Figure 5H, the capsule cells and capsule ECM cover the ends of the muscle spindle as if it was not attached to the tendons. The authors should be more specific.

–The legends of the supplementary tables are not provided

---

## [Author Response]

Reviewer #1 (Recommendations for the authors):– Please remember to add page and line numbers in your future manuscripts, it makes life much easier!

We thank the reviewer for this comment, following which we have added page and line numbers to our manuscript.

– In the introduction, third paragraph: the recent work of Lallemend and colleagues (Wu et al., Nat. Comm. 2021) needs to be cited along with "(Oliver et al., 2021; Wu et al., 2019)".

We apologize for this mix-up; we have corrected the reference.

– Please add in the manuscript how many spindles were collected (and from how many mice) for transcriptomics and proteomics analysis.

We thank the reviewer for this comment. We updated the Methods section, adding the number of spindles (~20) that were collected from each deep master sample.

– Results section, second paragraph: markers for intrafusal fibres, proprioceptive-SN and extracellular matrix are discussed but not for γ-MN. It would be great to add a sentence here about the most relevant that popped out in the screen.

We thank the reviewer for this comment. We have added a γ motor neuron marker to the paragraph (line 123) and to Figure 1C.

– "The 40 proteins that were differentially expressed in spindle vs muscle samples were grouped into five distinct clusters (Figure 2B)." In Figure 2B it says that "missing data are in white" not sure how to interpret it and there are many white boxes. Do the white boxes mean that the protein is not detected in the sample? or that there is something wrong with analysis? either way, how can you be sure this is a biological difference or just a technical issue and therefore validate the comparison? This point should be made clear.

We agree with the reviewer that the phrasing was confusing. For clarity, the words ‘missing data’ were replaced with ‘proteins not detected’.

– Figure 3BIII. How do you know apart from polar position that those fibers are γ-MN? it would be great to add a marker, VAChT?

We agree with the reviewer that adding a marker such as VAChT would confirm our identification of γ – MN innervation of the muscle spindle. Unfortunately, the suggested co-labeling with VAChT is challenging, because both the anti-ATP1a3 and the commonly used anti-VAChT antibodies are produced in rabbit. We tried using a goat antibody for VAChT and got no staining, even in neuromuscular junctions.

We concluded that the fibers in 3BIII are γ – MN based on three points: the anatomical location of innervation, on the polar ends of the intrafusal fibers; the unique morphology of these fibers, which is distinct from those of Ia and II innervations in the center of the spindle; and, the previously shown expression of ATP1a3 in γ – MN (Edwards et al., 2013).

Reviewer #2 (Recommendations for the authors):Overall, the data presented and the conclusions are very significant and based on well-conducted experiments. If the authors have the space, it would also be relevant to mention the major findings in the literature regarding the establishment of the muscle spindle by myogenic regulatory factors (MRFs of the MyoD family).

We agree with the reviewer that the involvement of MRFs in muscle spindle development is a fascinating subject. Given that muscle spindle differentiation occurs only after the generation of myotubes, MRFs mediate the initial myogenic stages of intra- and extrafusal fibers in a similar way. However, to our knowledge, this question has yet to be examined directly. Nevertheless, following this comment, we have added a description of spindle development (lines 50-54).

Do extrafusal and intrafusal formation rely on the same mechanisms with PAX7+ myogenic stem cells enabling postnatal muscle spindle growth?

The involvement of Pax7+ stem cells in postnatal muscle spindle growth is a very interesting question. Indeed, Pax3 and Pax7 are expressed by muscle spindle satellite cells (Kirkpatrick, et al., 2008; Kirkpatrick, et al., 2009), and depletion of Pax7-expressing cells results in atrophy of intrafusal fibers, thickening of muscle spindle-associated extracellular matrix and loss of motor coordination (Jackson et al., 2015). These findings indicate that satellite cells and, possibly, Pax7 have an unexplored role in adult muscle spindle function. We believe that the tools we provide in this manuscript will enable future studies of the effect of satellite cells on muscle spindle development and function.

Are myotendinous junctions also required at the ends of the muscle spindles to anchor these spindles in the tendons?

In our understanding, the myotendinous junctions are not required at the ends of muscle spindles. We and others observed that many spindle poles end well within the belly of a muscle (Banks et al., 2009; Banks et al., 2015). Author response image 1 shows the intrafusal fibers (marked by Myh7b) ending within the muscle belly (arrows), which raises the question of the mechanism that anchors these fibers.

**Author response image 1. sa2fig1:** The anchoring of muscle spindles. Confocal images of whole-mount EDL muscle from adult mice stained for Myh7b (green), to mark bag 2 intrafusal fiber, and ATP1a3 (magenta). The boxed area, which is enlarged below, demarcates an intrafusal fiber ending within the muscle belly. Arrows indicate the ends of the bag 2 fiber, dashed line indicates the end of the muscle. Scale bars represent 50 μm.

Main concerns– The authors in Figure 4 show that Myl2 is not detected: is a sarcomeric organization already established in the muscle spindle at this stage of development? This could be assessed by phalloidin staining for example and give us information on the potential biphasic specialization of this structure: a first common stage with extrafusal myofibers, then a second phase where the identity of intrafusal myofibers would be activated.

We thank the reviewer for this insight. Indeed, at early stages, Myl2 is not detected but the sarcomeric organization is already established. In order to better visualize this organization, we added a new Figure 4 —figure supplement 2, which shows phalloidin staining at various stages.

– In Figure S5, the ends of the fibers have been cut off. If the authors have pictures of these ends, they should provide them: what do the myotendinous junctions of the muscle spindles look like?

We thank the reviewer for this comment, and we have changed Figure 3 —figure supplement 2D to include the ends of the fibers as well.

In Figure 5, the ends of the myofibers are not shown, why? In Figure 5H, the capsule cells and capsule ECM cover the ends of the muscle spindle as if it was not attached to the tendons. The authors should be more specific.

We thank the reviewer for this comment and we changed Figure 5 accordingly. We have added the ends of the myofibers in E and we corrected the scheme in H.

–The legends of the supplementary tables are not provided

We have added supplementary table legends.